# UNSUPERVISED TEMPERATURE SCALING: ROBUST POST-PROCESSING CALIBRATION FOR DOMAIN SHIFT

## ABSTRACT

The uncertainty estimation is critical in real-world decision making applications, especially when distributional shift between the training and test data are prevalent. Many calibration methods in the literature have been proposed to improve the predictive uncertainty of DNNs which are generally not well-calibrated. However, none of them is specifically designed to work properly under domain shift condition. In this paper, we propose Unsupervised Temperature Scaling (UTS) as a robust calibration method to domain shift. It exploits unlabeled test samples instead of the training one to adjust the uncertainty prediction of deep models towards the test distribution. UTS utilizes a novel loss function, weighted NLL, which allows unsupervised calibration. We evaluate UTS on a wide range of model-datasets to show the possibility of calibration without labels and demonstrate the robustness of UTS compared to other methods (e.g., TS, MC-dropout, SVI, ensembles) in shifted domains.

## 1 INTRODUCTION

The predictive distributions provided by Deep Neural Networks (DNNs) have been increasingly used for decision-support systems, for applications ranging from medical diagnoses assistance (Esteva et al., 2017) to self-driving cars (Bojarski et al., 2016). In DNNs, the predictive distributions usually corresponds to the output of a softmax layer, which is typically interpreted as the confidence over the different classes. The i.i.d hypothesis made in learning usually assumes that the data distributions over the classes are the same at learning and inference time. However, in real-world applications, the distribution of data at inference time (i.e., the test data) may shift and actually be different from the original training distribution – corresponding to distribution shift in representation of data which we refer that as domain shift. For instance, in image classification problem, domain shift happens when the test images are different in illumination, view point, resolution, background or intensity noise from the training set. However, they are the same classification problem with the same objects occurance rate. Arguably, building DNNs that are robust to the domain shift problem is necessary for its safe deployment in decision-making systems. Dealing with this, predictive uncertainty is the key to obtain a meaningful estimation for practitioners to know when prediction accuracy is degrading and allows a system to abstain from making decisions due to low confidence.

The predicted uncertainty in DNNs usually are not calibrated with a tendency to be overconfident. Many probabilistic and post-processing calibration methods have been proposed under i.i.d assumptions to adjust the certainty of DNNs. In recent studies (Ovadia et al., 2019; Maddox et al., 2019), uncertainty under domain shift condition gets more attention and the common calibration methods have been assessed regarding to the domain shift, although they are not designed to be robust under such condition. In this paper, for the first time, we specifically focused on calibration for the domain shift in image classification. We show post-processing calibration approaches that use Negative Log Likelihood (NLL) as the calibration loss like Temperature Scaling (TS) (Guo et al., 2017) may become robust to the domain shift problem if they calibrate the model using the test samples. However, they need labels of the samples to apply calibration. Labeling the test samples even for the small set is not always an easy task and need human experts effort which can be accompanied with the labeling noise and huge time burden. Neuron cells classification taken by electron microscope (Ostroff & Zeng, 2015), pathology images (Khosravi et al., 2018) and skin disease classification (Kolkur et al.,

2018) are three examples of applications that have expensive labeling procedure with high risk of labeling noise that need senior experts to label them.

In this work, we propose a new approach called Unsupervised Temperature Scaling (UTS) with similar framework of TS and using unlabeled test samples for calibrating the pre-trained model. This novel idea brings the chance of robust calibration to the domain shift. Possibility of using the test samples to calibrate, makes UTS a proper solution not only for domain shift but also for many practical calibration problems like calibrating off-the-shelf-models. More specifically, UTS is proposed with following contributions and foreseen impacts:

- **Unsupervised post-processing calibration:** UTS brings a new look to NLL loss function which is used as the calibration loss in several post-processing methods. UTS approximates a weight function to estimate the per class distribution of data to compute NLL. This new way of computing NLL makes it independent of the labels. In addition, computing weighted NLL has the same order of time and memory complexity as the classic NLL, without any additional hyper-parameter to fine-tune.

- **Robustness to the domain shift**: UTS is a robust calibration solution to shifted domains. It adjusts the the model uncertainty based on the test and not the training domain. Therefore by change of distribution in the test domain, UTS can follow the distribution shift easily.

- **Calibration of off-the-shelf models**: The pre-trained models for classification tasks are usually only trained to achieve higher accuracy rate without paying attention to the predictive uncertainty. In fact, many of them are released without the training data that removes the possibility of retraining them to calibrate, like available Pytorch pre-trained models. UTS brings a chance to use these models for decision-making applications by calibrating them on a test data without need of labeling them.

## 2 RELATED WORK

Calibration of predictive uncertainty for DNNs are widely investigated in recent literature. Calibration methods can be categorized in two groups: probabilistic or post-processing approaches.

**Probabilistic approaches** refer to methods that use Bayesian theory (Bernardo & Smith, 2009) for estimating the conditional distribution of data. As the exact Bayesian inference is not practical, a variety of approximation are proposed such as Laplace approximation (MacKay, 1992; Ritter et al., 2018b;a; Kirkpatrick et al., 2017), Variational Bayesian methods (Molchanov et al., 2017; Louizos & Welling, 2017; Blundell et al., 2015; Louizos & Welling, 2016; Wen et al., 2018) and Monte Carlo Markov Chains (MCMC) (Neal, 2012; Balan et al., 2015; Chen et al., 2014) to make Bayesian deep networks tractable. MC-dropout (Gal & Ghahramani, 2016) replaces complicated sampling with simple dropout in training and test phases, which has been shown to approximate Variational Bayesian inference. Ensemble of DNNs (Lakshminarayanan et al. (2017)) is another straightforward probabilistic approach that can achieve better calibrated results than MC-dropout. This approach is appropriate for parallel computing, with multiple DNNs running at the same time. However, keeping the models in the memory during the test time brings high memory complexity.

**Post-processing approaches** are much less complex, albeit less accurate compared to probabilistic calibration. In post-processing approaches, the main idea is to decrease the miscalibration of the network by minimizing a calibration loss (Gneiting & Raftery, 2007) such as NLL. In order to train the neural network, NLL is used to simultaneously increase accuracy and decrease miscalibration. However, it easily gets overfitted to confidence and makes the network overconfident (Guo et al., 2017). Post-processing approaches like TS, Platt-Scaling (Platt et al., 1999), Histogram Binning (Zadrozny & Elkan, 2001), Isotonic Regression (Zadrozny & Elkan, 2002), and Bayesian Binning into Quantiles (Naeini et al., 2015) fine-tunes the softmax layer by keeping the DNNs' weights unchanged. They do not need to retrain the deep network from scratch and they only need to find the best parameter of softmax softening function by minimizing a calibration loss (like NLL) on a small validation set. Temperature Scaling is the state-of-the-art among the post-processing approaches which uses NLL as the loss function. It only uses one parameter $T$ to rescale the logit layer and soften the softmax output. Therefore with keeping the accuracy unchanged it can calibrate the model with the minimum time and memory complexity. These features leads us to focus on TS and try to propose a robust post-processing solution for domain shift based on TS framework.

**Robustness to the domain shift**: Previously, the results of calibrated model were also reported for different domains such as Out-Of-Distribution (OOD) and Adversaries (Lakshminarayanan et al., 2017; Ritter et al., 2018b) to show the model is uncertain about what it does not learn before. Recently, people get into importance of domain shift problem in calibration and assess how the calibrated methods would behave under domain shift condition (Ovadia et al., 2019; Maddox et al., 2019). Domain shift concept is different from adversaries and OOD. In the case of OOD, training and test domains are completely different in task distributions and in the case of adversaries the distribution shifts between the training and test is made with the goal of fooling the classifier. In domain shift, the training and test domains are distributionally different but related. The relation between two domains can be used as the prior knowledge to help improving the accuracy or having better calibration. In the literature of calibration, to the best of our knowledge, there is no work that specifically designed to calibrate the model considering domain shift assumptions. In this paper we will focus on Covariate shift as the most famous domain shift setting in image classification and propose UTS as a robust calibration method accordingly.

## 3 PRELIMINARIES

In this section, we define the domain shift and calibration setup to clarify UTS objectives. Then, we explain why NLL can be used as a calibration loss and when optimizing NLL will lead to a calibrated model toward the domain shift settings. Finally, we bring a deep analysis of the post-processing method TS which uses NLL as a calibration loss. We show TS can be a robust calibration solution to the domain shift if it uses labeled samples from the test domain to apply calibration. We discuss TS sensitivity to the labels of samples as the preliminaries to propose UTS method in the next section.

### 3.1 PROBLEM SETUP

Considering the domain shift assumptions, the goal of calibration in this work is to improve uncertainty estimation of a pre-trained model for different shifted domains. In this setting, $q_s(\mathbf{x}, y)$ is considered as the ground-truth distribution of the **source domain** and $q_t(\mathbf{x}, y)$ is considered as the ground-truth distribution of the **target domain** where $\mathbf{x} \sim \mathcal{X} \in \mathbb{R}^d$ and $y \in \{1, 2, \ldots, K\}$. **In the setting of domain shift, the source and target domains have different but related distributions**. The relation between the domains is defined by Covariate Shift assumption (Adel & Wong (2015)) which is: the data distributions $q_s(\mathbf{x}, y) \neq q_t(\mathbf{x}, y)$ where the conditional distribution $q_s(y|\mathbf{x}) = q_t(y|\mathbf{x})$, and the task and marginal distributions $q_s(y) = q_t(y)$ and $q_s(\mathbf{x}) \neq q_t(\mathbf{x})$, respectively. Let $d(\mathbf{x}) = \{S_y(\mathbf{x}), \hat{y}\}$ denotes the pre-trained model in which $\hat{y}$ is the class prediction and $S_y(\mathbf{x})$ is its associated confidence. In domain shift setting, for deep neural networks, model $d(\cdot)$ is a DNN trained on the source domain and would be tested on the target domain. In this setting, $S_y(\mathbf{x})$ is the output of the softmax layer which is calibrated when $S_y(\mathbf{x}) = q_t(y|\mathbf{x})$.

Miscalibration of DNNs can be explained in different ways. Temperature Scaling models the miscalibration as the rescaled logit layers by scaling factor $T^*$. TS objective is to find $T^*$ value to rescale the logit layer back and makes the model calibrated. More specifically, the calibrated output of TS is defined as $S_y(\mathbf{x}; T^*) = \exp(\frac{f_y(\mathbf{x})}{T^*}) / \sum_{j=1}^{K} \exp(\frac{f_j(\mathbf{x})}{T^*})$ where $\mathbf{f}(\mathbf{x}) = [f_1(\mathbf{x}), f_2(\mathbf{x}), \ldots, f_k(\mathbf{x})]^\top$ is the logit layer of model $d(\mathbf{x})$. In this paper, considering the same definition of miscalibration as TS, we propose UTS under domain shift condition. **UTS objective** is to find the scaling factor $T^*$ that $S_y(\mathbf{x}; T^*) = q_t(y|\mathbf{x})$, given that we have access to the source pre-trained model $d(\cdot)$, unlabeled calibration set $\mathbb{C} = \{\mathbf{x}_i\}_{i=1}^{L} \sim q_t(\mathbf{x})$ and the known task distribution $q_s(y)$.

### 3.2 ROBUSTNESS TO DOMAIN SHIFT WITH NLL LOSS FUNCTION

To calibrate a model, first we need to evaluate the quality of predicted uncertainty of the model. Evaluating the quality of predictive uncertainty is challenging, as the ground-truth of the uncertainty estimate is usually not available. Accordingly, scoring rules are defined to measure the quality of predictive uncertainty (Gneiting & Raftery (2007)). Scoring rules are numerical scores that rank the distribution prediction $p_\theta(y|\mathbf{x})$ by giving lower score to better prediction of true distribution $q(y|\mathbf{x})$. Let a scoring rule be a function $R(p_\theta, (\mathbf{x}, y))$ that evaluates the quality of the predictive distribution $p_\theta(y|\mathbf{x})$ based on the samples $(\mathbf{x}, y) \sim q(\mathbf{x}, y)$ where $q(\mathbf{x}, y)$ is the true distribution of the data.

The expected scoring rule is defined by $R(p_\theta, q) = \int q(\mathbf{x}, y) R(p_\theta, (\mathbf{x}, y)) dy d\mathbf{x}$. A **proper scoring rule** function is one where $R(p_\theta, q) \leq R(q, q)$ with equality if and only if $p_\theta(y|\mathbf{x}) = q(y|\mathbf{x})$ for all samples.

Negative Log Likelihood (NLL) is a proper scoring rule based on Gibbs inequality i.e., $R(p_\theta, q) = \mathbb{E}_{q(\mathbf{x})} q(y|\mathbf{x}) \log p_\theta(y|\mathbf{x}) \leq \mathbb{E}_{q(\mathbf{x})} q(y|\mathbf{x}) \log q(y|\mathbf{x})$. Therefore minimizing NLL w.r.t $\theta$ on the samples generated from $q(\mathbf{x}, y)$ distribution, leads to $p_\theta(y|\mathbf{x}) \to q(y|\mathbf{x})$. Under the domain shift assumption, as the training and test domains have different distributions, the final goal of calibration is $p_\theta(y|\mathbf{x}) = q_t(y|\mathbf{x})$. In the case of using NLL as the loss function, if we minimize NLL on the samples that are generated from the test domain, we will have $p_\theta(y|\mathbf{x}) \to q_t(y|\mathbf{x})$ and makes the method robust to the domain shift. One of the post-processing methods that uses NLL as the loss function is TS. Therefore, TS has the ability to get robust to the domain shift.

### 3.3 TEMPERATURE SCALING ANALYSIS

TS (Guo et al. (2017)) is the state-of-the-art post-processing approach which rescales the logit layer of a deep model by parameter $T$ that is called temperature. TS is used to soften the output of the softmax layer and makes it more calibrated. The best value of $T$ will be obtained by minimizing NLL loss function (explained in Sec.(3.2), why minimizing NLL leads to more calibrated model) respecting to $T$ conditioned by $T > 0$ on the calibration set as defined in Eq. (1):

$$
T_{TS}^* = \arg\min_T \overbrace{\left( -\sum_{i=1}^L \log \left( S_{y_i}(\mathbf{x}_i; T) \right) \right)}^{\text{NLL}} \quad s.t : T > 0, \quad \{\mathbf{x}_i, y_i\}_{i=1}^L \in \mathbb{C} \sim q(\mathbf{x}, y), \tag{1}
$$

where $S_{y_i}(\mathbf{x}_i; T) = \exp(\frac{\mathrm{f}_{y_i}(\mathbf{x}_i)}{T}) / \sum_{j=1}^K \exp(\frac{\mathrm{f}_j(\mathbf{x}_i)}{T})$, is the softed version of softmax by applying parameter $T$ on the logit layer $\mathrm{f}_j(\mathbf{x})$. TS has the minimum time and memory complexity with order of $\mathcal{O}(1)$ among calibration approaches as it only optimizes one parameter $T$ on small labeled calibration set. Having only one parameter helps TS not only to be efficient and practical but also not to get overfitted to NLL loss function when it is optimized on small calibration set. TS previously is applied for calibration (Guo et al. (2017)), distilling the knowledge (Hinton et al. (2015)) and enhancing the output of DNNs for better discrimination between the in and out distribution samples (Liang et al. (2017)). TS models the uncalibration as the rescaling factor in the logit layer. By computing the derivative of NLL respecting to $T$ in Eq. (1), and find the minimum, we will have:

$$
\sum_{i=1}^L \mathrm{f}_{y_i}(\mathbf{x}_i) = \sum_{i=1}^L \sum_{k=1}^K \mathrm{f}_k S_k(\mathbf{x}_i; T_{TS}^*). \tag{2}
$$

It shows regarding to the true label of the samples, TS selects the $T$ value which maximizes the $S_k(\mathbf{x}_i, T)$ for $k = y_i$ and minimize $S_k(\mathbf{x}_i, T)$ for all the other $k \neq y_i$. Therefore for correctly classified samples that $y_i = \arg\max_y S_y(\mathbf{x}_i)$, $T$ approaches 0 to increase the confidence of that class toward 1 and for the misclassified samples, $T$ goes toward $\infty$ to decrease the confidence of predicted label. The balance between the correctly classified and misclassified samples brings back the optimal value of $T$.

TS uses NLL as the loss function then by selecting the calibration set from the test domain instead of the training one, we can make TS robust to the domain shift problem. We refer to this approach as **TS-Target**. TS is highly dependent on the labels of the samples. however, Labeling the test samples is a challenging task. When the calibration set contains data with labeling noise or unlabeled samples, TS loses the balance to find the optimal $T_{TS}^*$ and cannot calibrate the network successfully. Later, in Sec. (5.3), we will show even a small portion of the noise has big distortion in TS results. This brings the idea of Unsupervised Temperature Scaling which makes TS independent of labeled data and robust to the domain shift.

## 4 UNSUPERVISED TEMPERATURE SCALING

Considering the assumptions in Sec. (3.1), its main objective is to find $T_{UTS}^*$ for the unlabeled calibration set. UTS the same as TS uses NLL as the proper scoring rule to minimize the calibration

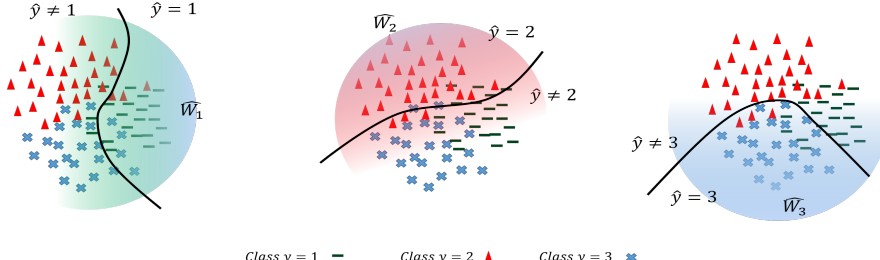

Figure 1: A color view of the weight function in a three class classification problem. For the samples which are classified as class $\hat{y} = k$ and for the samples that $\hat{y} \neq k$ but located near to the decision boundary, the $\hat{W}_k(\mathbf{x}, w^*) = 1$ which is shown with darker hue. For the samples that $\hat{y} \neq k$ and are far from the decision boundary $\hat{W}_k(\mathbf{x}, w^*) \to 0$, which is with lighter hue. The decision boundary is illustrated by the black line.

gap and finds the optimal $T$ value. The first step of using NLL in UTS is to make NLL independent of labeled data. Considering NLL loss function, we can rewrite it with focus on per class distribution, formalized as:

$$\text{NLL} = -\sum_{k=1}^{K} \sum_{(\mathbf{x}_i, y_i) \in q(\mathbf{x}, y=k)} \log \left( S_{y_i}(\mathbf{x}_i; T) \right) \quad , \tag{3}$$

In Eq. (3), NLL is the summation of $K$ different sample sets, generated from class distribution $q(\mathbf{x}, y = k)$ where $k \in \{1, 2, \ldots, K\}$. When the labels of the samples are available, they can be used as the guide to select the samples set for each class distribution. But in the absence of the labels, the main question is how to select the samples generated from $q_t(\mathbf{x}, y = k)$ and calculate NLL. To come along with this challenge, instead of selecting the samples by labels, UTS applies weights on the samples. The weight function $\hat{W}_k(\mathbf{x}; w^*)$ represents the probability that sample $\mathbf{x}$ is drawn from $q(\mathbf{x}, y = k)$. Later, in Sec. (4.1) we will give specific details of the weight function $\hat{W}_k(\mathbf{x}; w^*)$ and how to approximate it. By applying weights on the samples, the UTS loss function is defined as the Weighted NLL (WNLL) which is:

$$T^*_{UTS} = \arg\min_{T} \overbrace{\left( -\sum_{k=1}^{K} \sum_{i=1}^{L} \hat{W}_k(\mathbf{x}_i; w^*) \log \left( S_k(\mathbf{x}_i; T) \right) \right)}^{\text{WNLL}} \quad s.t: T > 0, \quad \{\mathbf{x}_i\}_{i=1}^{L} \in \mathbb{C} \sim q(\mathbf{x}) \tag{4}$$

### 4.1 Weight Function $\hat{W}_k(\cdot; \cdot)$

We start the discussion about the weight function by introducing a fact from the Bayes rule (Jin et al. (2017)):

$$q(\mathbf{x}, y = k) = \frac{q(y = k|\mathbf{x})}{q(y \neq k|\mathbf{x})} q(\mathbf{x}, y \neq k) \tag{5}$$

When a sample has a true label of $y = k$ it is drawn from distribution $q(\mathbf{x}, y = k)$. However, Eq.(5) shows that even samples with true label of $y \neq k$ can be used as the samples drawn from the distribution $q(\mathbf{x}, y = k)$ by applying weight of $q(y = k|\mathbf{x})/q(y \neq k|\mathbf{x})$. Therefore, we can simply use the weight of 1 for the samples with true label of $y = k$ and the weight of $q(y = k|\mathbf{x})/q(y \neq k|\mathbf{x})$ for the samples $y \neq k$ to change all the samples in the calibration set to the samples drawn from $q(\mathbf{x}, y = k)$. Therefore, we define the weight function as Eq.(6):

$$W_k(\mathbf{x}_i) = \begin{cases} 1, & \text{if } \mathbf{x}_i \sim q(\mathbf{x}, y = k) \\ \frac{q(y=k|\mathbf{x}_i)}{1-q(y=k|\mathbf{x}_i)}, & \text{otherwise.} \end{cases} \tag{6}$$

---

**Algorithm 1:** Unsupervised Temperature Scaling

---

**Require:** $q_s(y)$: task distribution
**Require:** $d(\cdot)$: the pre-trained model
**Require:** $\mathbb{C} \sim q_t(\mathbf{x})$: unlabeled calibration set derived from test domain
1: Find the optimal $w^* = \arg\min_w (\sum_{k=1}^{K} \sum_{i=1}^{L} \hat{W}_k(\mathbf{x}_i; w) - q(y = k))^2$,
2: Find the optimal $T_{UTS}^* = \arg\min_T \left( -\sum_{k=1}^{K} \sum_{i=1}^{L} \hat{W}_k(\mathbf{x}_i; w^*) \log (S_k(\mathbf{x}_i; T)) \right)$
3: Calibrate softmax output of model $d(\cdot)$ by: $S_y(\mathbf{x}; T_{UTS}^*) = \exp(\frac{f_y(\mathbf{x})}{T_{UTS}^*}) / \sum_{j=1}^{K} \exp(\frac{f_j(\mathbf{x})}{T_{UTS}^*})$ .

---

To compute $W_k(\cdot)$ for the samples, we need the ground-truth distribution $q(\mathbf{x}, y = k)$ that in UTS setting is not available. However we can approximate it empirically referring to UTS assumptions (Sec. (3.1)).

**Proposition 1**: *Let $d(\cdot)$ be a model which gets miscalibrated with rescaled logit layer by factor $w^*$. Then, with known $q_t(y)$, the empirical approximation of $W_k(\cdot)$ is equal $\hat{W}_k(\cdot; \cdot)$ which is defined as*:

$$\hat{W}_k(\mathbf{x}_i, w^*) = \begin{cases} 1, & \text{if} \quad \hat{y}_i = k \\ 1/\exp(\frac{1}{w^*} \log(S_{y=k}(\mathbf{x}_i; \frac{1}{w^*})^{-1} - 1)), & \text{otherwise.} \end{cases} \tag{7}$$

where $w^*$ is:

$$w^* = \arg\min_w \left( \sum_{k=1}^{K} \sum_{i=1}^{L} \hat{W}_k(\mathbf{x}_i; w) - q_t(y = k) \right)^2, \quad \{\mathbf{x}_i\}_{i=1}^{L} \in \mathbb{C} \sim q_t(\mathbf{x}) \tag{8}$$

Proposition 1 valid for the domain shift setting with Covariate shift assumption and also for the case that there is no distribution shift between the training and test datasets. The validity of Proposition 1 for both settings are provided in Appendix A. Fig. 1 illustrates a schematic view of the weight function $\hat{W}_k(\cdot; \cdot)$ in the feature space. The color hue is correlated with the weight values. For the samples that are classified as $\hat{y} = k$ and for the samples with $\hat{y} \neq k$ located near to the decision boundary, the weight is equal to 1. The weight would decrease as the samples fall further from the decision boundary which shows they are less probable to be drawn from distribution $q(\mathbf{x}, y = k)$.

**Time Complexity of UTS:** Computing $\hat{W}_k(\cdot; \cdot)$ is a one parameter optimization which has the time complexity of $\mathcal{O}(1)$. After approximating the weight function $\hat{W}_k(\cdot; \cdot)$, UTS minimizes WNLL (Eq.(4)) with the same time complexity to find the optimal $T_{UTS}^*$ which leads to have UTS with the total time complexity of $\mathcal{O}(1)$. Algorithm 1 summarizes UTS.

**Validity of UTS in Practice:** UTS is valid when there is no domain shift or when the there is Covariate Shift between domains. When the test and training datasets are different in representation but keeps the same proportions of each class occurrence, it is categorized as Covariate shift assumption. In many applications like medical image classifications, the probability of happening a class of object is staying the same during the training and test phases which means $q_s(y) = q_t(y)$ but the illumination, capturing noise, resolution, and image size and viewpoint can vary between two domains which means $q_s(\mathbf{x}) \neq q_t(\mathbf{x})$. Therefore, in classification problems with Covariate Shift assumption or without any shift, UTS only needs to calculate empirically the number of occurrence of each class to the total number of samples in the training set and use it as $q_t(y)$ to calibrate the model.

## 5 EXPERIMENTS

We conduct the experiments to analyze the behavior of UTS in comparison to the other methods for two different calibration scenarios. First, we compare UTS with several post processing methods that use NLL as the loss function in the experiment with **the same training and test domain distributions**. This experiment is designed to be a proof of concept to show that weighted NLL of UTS can indeed calibrate the model without accessing the labels. Second, in order to show the success of UTS in calibrating the model under domain shift condition, we compare UTS, TS and

Table 1: The results of NLL↓ for UTS and other post-processing approaches with the same training and test domains. In all cases UTS can calibrate the model without labeled samples, however it does not achieve the best results. This experiment shows UTS can calibrate off-the-shelf-models with only test samples. The mean are reported in 20 independent runs.

| Dataset | Model | Uncalibrated | TS | UTS | Vector Scaling | Matrix Scaling |
|---------|-------|--------------|-----|-----|----------------|----------------|
| Birds | ResNet50 | 0.9383 | **0.9287** | 0.9313 | 0.9355 | 7.1423 |
| MNIST | Lenet-5 | 0.1044 | **0.0243** | 0.0429 | 0.0988 | 0.1187 |
| SVHN | ResNet110 | 0.2107 | **0.1534** | 0.1566 | 0.1936 | 0.2070 |
| SVHN | DenseNet100 | 0.1803 | **0.1608** | 0.1735 | 0.1687 | 0.1754 |
| CIFAR10 | DenseNet40 | 0.2895 | **0.2221** | 0.2848 | 0.2912 | 0.2883 |
| CIFAR10 | DenseNet100 | 0.1973 | **0.1559** | 0.1635 | 0.2006 | 0.1965 |
| CIFAR10 | ResNet110 | 0.3134 | **0.2069** | 0.2341 | 0.2907 | 0.3103 |
| CIFAR10 | VGG16 | 0.2608 | **0.2036** | 0.2099 | 0.2503 | 0.2573 |
| CIFAR100 | DenseNet40 | 1.0964 | **1.0064** | 1.0217 | 1.2876 | 1.4914 |
| CIFAR100 | DenseNet100 | 1.1285 | **0.8751** | 0.8889 | 1.2470 | 1.2761 |
| CIFAR100 | ResNet110 | 1.2442 | **1.0466** | 1.0636 | 1.3869 | 1.5358 |
| CIFAR100 | WideResNet40-4 | 0.8751 | **0.8192** | 0.8439 | 0.8757 | 0.8609 |

3 more probabilistic approaches for **the training and test domains with different distributions**. We also study the results of **TS-Target** that is a TS which selects the calibration set from the target (test) domain. TS-Target has the most accurate uncertainty prediction among all other baselines for the shifted domain distributions. However, we will show in the third section of Experiments part, it suffers from the labeling noise which justifies our try to make TS unsupervised.

## 5.1 Calibration with the Same Training and Test Domains

Here we consider the training and the test domains have the same distribution. Our goal is to show UTS can calibrate the models without labels in the case of no domain shift.

**Experiment Setup** We compare UTS with several post-processing baselines which are Temperature Scaling (TS) (Guo et al. (2017)) and Matrix and Vector Scaling (Platt et al. (1999)) on a wide range of different state-of-the-art deep convolutional networks with variations in depth which are ResNet (He et al. (2016)), WideResNet (Zagoruyko & Komodakis (2016)), DenseNet (Iandola et al. (2014)), LeNet (LeCun et al. (1998a)), and VGG (Simonyan & Zisserman (2014)). We test the methods on different datasets such as CIFAR-10 and CIFAR-100 (Krizhevsky & Hinton (2009)), SVHN (Netzer et al. (2011)), MNIST (LeCun et al. (1998b)), and Calthec-UCSD Birds (Wah et al. (2011)). We use all the data pre-processing, training procedures and hyper-parameters tuning for each dataset as described in each mentioned reference. To setup the calibration set, we randomly select 20% of the test dataset. Then, we consider the rest to be evaluated as a test set. We repeat each experiment 20 times independently and report the mean of NLL as a calibration metric. More explanation of experiment setup and the baselines, datasets and calibration metrics are provided in Appendix B.

**Results** In Table 1, the calibration results based on NLL are compared between TS and UTS, which have only one parameter for fine-tuning the softmax output layer, and Matrix and Vector scaling, which apply a linear function on logit layers. In all cases, TS calibrates the network better than all the other methods. Although the results of UTS are not better than TS, UTS shows improvement in calibration for all dataset-models. It means the weighted NLL as the approximation of NLL with unlabeled samples can work properly to calibrate the model even though it is not as accurate as the NLL with access to labels. Although Matrix and Vector Scaling can define more complex functions to soften the softmax layer, they suffer from over-fitting w.r.t the validation set in confidence. We also provide the complete results with mean and standard deviation for accuracy as well as other standard calibration metrics (NLL, ECE and Brier Score) in Appendix C.1. The explanation about the ECE and Brier are given in Appendix B.3.

## 5.2 Calibration Under Domain Shift Setting

In this section, we divide the experiments into two parts. First, we compare UTS to the uncalibrated model (UNC), TS and TS-Target on different domain shift scenarios. The goal of this experiment is to compare the gap of calibration between UTS and TS-Target that can be considered as the ground-

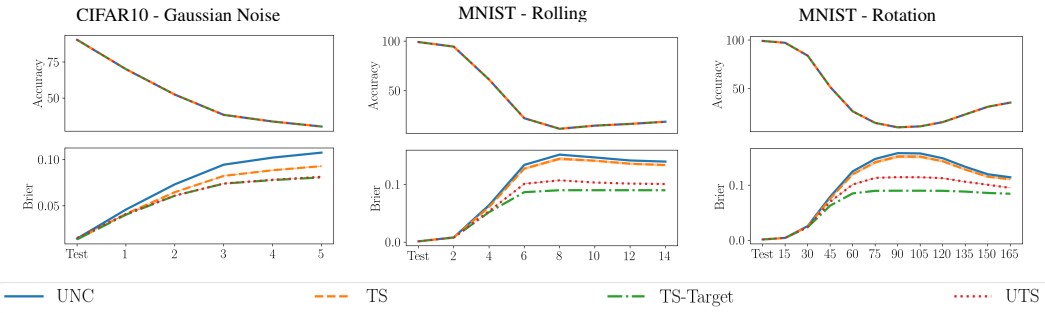

Figure 2: Results show accuracy and Brier Score↓ in different degree of shifting for MNIST and CIFAR10 datasets. Comparing TS and TS-Target results shows TS-Target is more robust to domain shift than TS. UTS with small gap is following TS-Target that is the labeled version of UTS and can be considered as the ground-truth for it. The pre-trained models are trained on MNIST and CIFAR10 datasets and tested on different degree of shifted MNIST and CIFAR10 datasets to model domain shift.

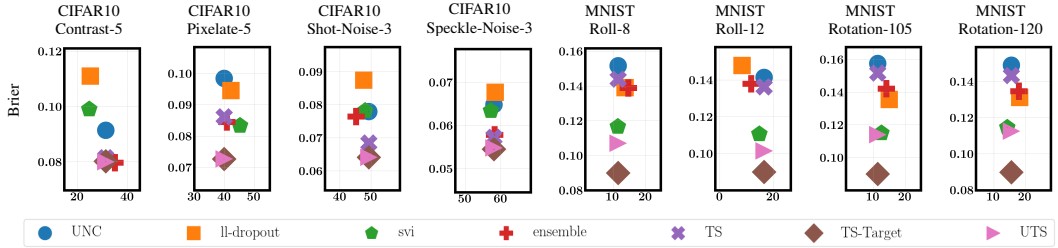

Figure 3: Comparison between different post-processing and probabilistic approaches under domain shift. The results are reported for Brier Score↓ vs Accuracy. TS-Target has the most calibrated output comparing to the other probabilistic approaches. UTS after TS-Target achieves the best results which shows comparing to the other probabilistic approaches, it is robust to the domain shift. The models are trained on MNIST and CIFAR10 later during the test different degrees of shifts applied on them to make the domain shift

truth for UTS when the labels are available. Later, we also evaluate the robustness of UTS, which was specifically designed to domain shift, and several probabilistic approaches, which only consider the case of calibration for the same distribution setting. The goal of the experiment is to show that UTS can be indeed robust for different shifting domain scenarios.

**Experiment Setup** We follow the same experimental setup as Sec (5.1) but with different domain shift assumptions. We use the benchmark proposed specifically for domain shift problem in (Ovadia et al. (2019)). They model the distribution shift by applying different operations like rotation, translation (rolling), and with different severity levels of intensity corruptions proposed in (Hendrycks & Dietterich (2019)). In the first part of this section, we compare the result of the UTS to uncalibrated, TS and TS-Target on MNIST and CIFAR10 datasets applying rotation, pixel translation and Gaussian noise to the test domain. In the second part, we add more probabilistic baselines such as LL-Dropout(Gal & Ghahramani (2016)), SVI (Blundell et al. (2015)) and Ensemble (Lakshminarayanan et al. (2017)) with more variation of the domain shifts in the sense of corruption of intensities. Specific details of baselines and experiment can be found in Appendix B.1

**Results** As we can see in Fig. 2, the accuracy of the model degrades by the effect of domain shift. TS family approaches does not change the accuracy during the calibration therefore, all the methods have the same accuracy as the uncalibrated one. TS-Target has the same setting as TS with the difference that it uses the labeled calibration set from the target domain. Then, in ideal situation, UTS uncertainty prediction would reach the TS-Target performance. We can see that UTS is working better than uncalibrated model (UNC) and TS which uses the source data for calibration under the domain shift condition. The gap between UTS and TS-Target is interestingly small in the sense of Brier Score. More results for other domain shifts are provided in Appendix C.2. We also provide the analysis of UTS sensitivity to number of calibration samples in compared to TS and TS-Target in Appendix C.4 which shows UTS obtains stable results in decreasing the number of samples from 20% to 2.5% of test detest size.

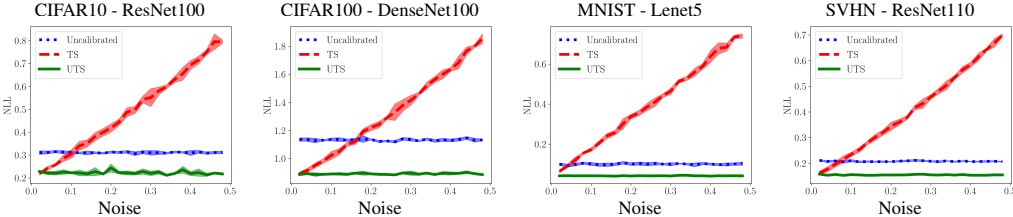

Figure 4: Sensitivity of TS approach to the labeling noise. Even with small percentage of noise in the labels of samples TS cannot calibrate the model. UTS is completely robust to the labeling noise as it is an unsupervised approach. Shaded regions represent std over 20 runs.

In Fig. 3 we compare probabilistic approaches to UTS, uncalibrated model (UNC), TS and TS-Target. As all the calibration metrics are dependent on the accuracy of the models, controlling the accuracy to have fair comparison between the methods is important. Otherwise, the better calibration can be as the result of having better accuracy and not the calibration itself. Accordingly, we apply the shifts to the datasets and check the accuracy of the UTS with other approaches, and select the domain shift settings that UTS accuracy is near to the others. As we can see for different combination of model and datasets, UTS can achieve better results than any other probabilistic approaches and has a small gap with TS-Target which achieves the best results. It shows that using the test samples to fine-tune the calibration toward test distribution can help the model to be robust to the domain shift problem. As mentioned before, labeling the test samples is not a trivial task, therefore, using directly TS-Target might not be possible in many cases which justifies the importance of an unsupervised approaches like UTS. In the next section, we show that for a weakly supervision of the test samples, TS-Target cannot be successful in calibration and it needs exact labeling of the test domain samples that might be impractical in many cases.

## 5.3 TS SENSITIVITY TO LABELING NOISE

When the labeled samples are available for calibration, TS shows the best results with and without domain shifts. In this section, we will investigate the sensitivity of TS with labeling noise. We apply different rates of random altering the labels only for the calibration set and evaluate the calibration success of TS, UTS and uncalibrated methods accordingly. As we can see in Fig. 4, TS is extremely sensitive to the noise of labeling. Therefore, in order to have a successful calibration for TS in shifted domains, the exact labeling of test samples is essential which might be not feasible for many applications. UTS is robust to the labeling noise as it is an unsupervised calibration method and it can remove the challenge of labeling the test samples, completely. More results for more datasets-models are provided in Appendix C.3.

## 6 DISCUSSION AND FUTURE WORK

In this paper, we propose UTS as a robust unsupervised post-processing method to the domain shift calibration challenge. UTS is a member of TS family approaches which have low time and memory complexity, and can calibrate with few number of samples while preserving the accuracy intact. UTS utilizes a new calibration loss function, weighted NLL which is independent of the labels. The computational complexity of weighted NLL is in the same order of NLL which makes UTS a fast and practical calibration solution. Since UTS uses the test samples to adjust the uncertainty, we show it is robust to domain shift and can make off-the-shelf models calibrated when their training samples are not available anymore.

Recent studies (Maddox et al., 2019; Kumar et al., 2018) mentioned that using TS with probabilistic approaches can even improve the uncertainty prediction of already calibrated models. Therefore, we believe this work can be extended in the direction of combining UTS with such approaches in order to achieve more robust domain shift solutions. We also consider another direction of this work towards exploring UTS for more variant domain shift assumptions. In this paper, we study UTS only for Covariate Shift assumption, however, it can be extended to other shifting scenarios such as OOD and adversaries, in future.

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

## A  PROOF OF PROPOSITION 1

First we investigate the validity of Proposition 1 for the settings with no distribution shift.
**Proposition 1**: *let $W_k(\cdot)$ be a weight function defined as:*

$$W_k(\mathbf{x}_i) = \begin{cases} 1, & \text{if} \quad \mathbf{x}_i \sim q(\mathbf{x}, y = k) \\ \frac{q(y=k|\mathbf{x}_i)}{1-q(y=k|\mathbf{x}_i)}, & \text{otherwise.} \end{cases}$$

*and let $d(\cdot)$ be a model which gets miscalibrated with rescaled logit layer by factor $w^*$. Then, with known task distribution $q(y)$, the empirical approximation of $W_k(\cdot)$ is equal $\hat{W}_k(\cdot;\cdot)$ which is defined as:*

$$\hat{W}_k(\mathbf{x}_i, w^*) = \begin{cases} 1, & \text{if} \quad \hat{y}_i = k \\ 1/\exp(\frac{1}{w^*}\log(S_{y=k}(\mathbf{x}_i)^{-1} - 1)), & \text{otherwise.} \end{cases}$$

*where $w^*$ is:*

$$w^* = \arg\min_w \left( \sum_{k=1}^{K} \sum_{i=1}^{L} \hat{W}_k(\mathbf{x}_i; w) - q(y=k) \right)^2 \quad \mathbf{x}_i \in \mathbb{C} \sim q(\mathbf{x})$$

**Proof**: *For simplicity we split the proof into two parts. First, we show that for a known value of $w^*$, $\hat{W}_k(\cdot;\cdot)$ is the approximation $W_k(\cdot)$. In other words, $1/\exp(\frac{1}{w^*}\log(S_{y=k}(\mathbf{x})^{-1} - 1)) = q(y=k|\mathbf{x})/q(y\neq k|\mathbf{x})$:*

*The softmax output of an uncalibrated model $d(\cdot)$ with rescaled logit layer by a factor $w^*$, can be formulated as:*

$$S_{y=k}(\mathbf{x}) = \frac{\exp(w^* \mathbf{f}_k(x_i))}{\sum_{j=1}^{K} \exp(w^* \mathbf{f}_j(\mathbf{x}_i))}$$

$$S_{y\neq k}(\mathbf{x}) = 1 - S_{y=k}(\mathbf{x})$$

*Therefore calibrated output will be defined as:*

$$S_{y=k}(\mathbf{x}; w^*) = \frac{\exp(\mathbf{f}_k(x_i))}{\sum_{j=1}^{K} \exp(\mathbf{f}_j(\mathbf{x}_i))} = q(y=k|\mathbf{x})$$

$$S_{y\neq k}(\mathbf{x}, w^*) = 1 - S_{y=k}(\mathbf{x}, w^*) = q(y\neq k|\mathbf{x})$$

*Considering these definitions:*

$$\frac{1}{\exp\left(\frac{1}{w^*}\log(S_{y=k}(\mathbf{x})^{-1} - 1)\right)} = \frac{1}{\exp\left(\frac{1}{w^*}\log\left(\frac{\sum_{j=1}^{K}\exp(w^*\mathbf{f}_j(\mathbf{x}_i))-\exp(w^*\mathbf{f}_k(x_i))}{\exp(w^*\mathbf{f}_k(\mathbf{x}_i))}\right)\right)}$$

$$= \frac{1}{\exp\left(\log\left(\frac{\sum_{j=1}^{K}\exp(\mathbf{f}_j(x_i))-\exp(\mathbf{f}_k(x_i))}{\exp(\mathbf{f}_k(\mathbf{x}_i))}\right)\right)}$$

$$= \frac{1}{\left(\frac{\sum_{j=1}^{K}\exp(\mathbf{f}_j(\mathbf{x}_i))-\exp(\mathbf{f}_k(\mathbf{x}_i))}{\exp(\mathbf{f}_k(\mathbf{x}_i))}\right)}$$

$$= \frac{\exp(\mathbf{f}_k(\mathbf{x}_i))}{\left(\sum_{j=1}^{K}\exp(\mathbf{f}_j(\mathbf{x}_i))-\exp(\mathbf{f}_k(\mathbf{x}_i))\right)}$$

$$= \frac{\left(\frac{\exp(\mathbf{f}_k(\mathbf{x}_i))}{\sum_{j=1}^{K}\exp(\mathbf{f}_k(\mathbf{x}_i))}\right)}{\left(1 - \frac{\exp(\mathbf{f}_k(\mathbf{x}_i))}{\sum_{j=1}^{K}\exp(\mathbf{f}_j(\mathbf{x}_i))}\right)}$$

$$= \frac{S_{y=k}(\mathbf{x}; w^*)}{1 - S_{y=k}(\mathbf{x}; w^*)} = \frac{q(y=k|\mathbf{x})}{1 - q(y=k|\mathbf{x})} = \frac{q(y=k|\mathbf{x})}{q(y\neq k|\mathbf{x})}$$

*We consider $\hat{y}_i = k$ as the rough estimation of condition $\mathbf{x}_i \sim q(\mathbf{x}, y = k)$. The accuracy of the uncalibrated model is the same as calibrated one when it gets uncalibrated by rescaled logit layer. Therefore we can use the prediction output of uncalibrated model to have a rough estimation of the samples that $\mathbf{x} \sim q(\mathbf{x}, y = k)$.*

*Now in the second part, we show that $w^* = \arg\min_w \left( \sum_{k=1}^{K} \sum_{i=1}^{L} \hat{W}_k(\mathbf{x}_i; w) - q(y = k) \right)^2$ where $\mathbf{x}_i \in \mathbb{C} \sim q(\mathbf{x})$*

*Referring to $W_k(\cdot)$ definition, we can simply show:*

$$\int_x W_k(\mathbf{x}) q(\mathbf{x}, y) d\mathbf{x} = q(y = k)$$

*Which means $\mathbb{E}_{q(\mathbf{x}, y)}[W_k(\mathbf{x})] = q(y = k)$, and as $\hat{W}_k(\cdot; \cdot)$ is equal to $W_k(\cdot)$, empirically we can show $\sum_{\mathbf{x}_i^c \in \mathbb{C}} \hat{W}_k(\mathbf{x}_i; w^*) = q(y = k)$. In this problem setting, we assume $q(y)$ is known. Therefore, $w^*$ can be found easily by minimizing $\left( \sum_{k=1}^{K} \sum_{i=1}^{L} \hat{W}_k(\mathbf{x}_i; w) - q(y = k) \right)^2$.*

*Now we show the validity of Proposition 1 under Covariate Shift assumptions.*

**Corollary 1**: *Considering the same weight function $W_k(\cdot)$ defined in Proposition 1, let $d(\cdot)$ be a model which gets miscalibrated with rescaled logit layer by factor $w^*$ toward the target domain $q_t(\mathbf{x}, y)$. Assume covariate shift where $q_s(y|\mathbf{x}) = q_t(y|\mathbf{x})$, $q_s(y) = q_t(y)$ and $q_s(\mathbf{x}) \neq q_t(\mathbf{x})$. Then, with known $q_s(y)$, the empirical approximation of $W_k(\cdot)$ is equal $\hat{W}_k(\cdot; \cdot)$ which is defined as:*

$$\hat{W}_k(\mathbf{x}_i, w^*) = \begin{cases} 1, & \text{if } \hat{y}_i = k \\ 1/\exp(\frac{1}{w^*} \log(S_{y=k}(\mathbf{x}_i)^{-1} - 1)), & \text{otherwise.} \end{cases}$$

*where $w^*$ is:*

$$w^* = \arg\min_w \left( \sum_{k=1}^{K} \sum_{i=1}^{L} \hat{W}_k(\mathbf{x}_i; w) - q_s(y = k) \right)^2 \quad \mathbf{x}_i \in \mathbb{C} \sim q_t(\mathbf{x})$$

**Proof**: *The first part of the proof is exactly the same as Proposition 1 with considering that $w^*$ with new assumptions is the scaling factor that makes the model uncalibrated towards the target domain. Therefore, we will conclude:*

$$= \frac{S_{y=k}(\mathbf{x}; w^*)}{1 - S_{y=k}(\mathbf{x}; w^*)} = \frac{q_t(y = k|\mathbf{x})}{1 - q_t(y = k|\mathbf{x})} = \frac{q_t(y = k|\mathbf{x})}{q_t(y \neq k|\mathbf{x})}$$

*For the second part of the proof, we have:*

$$\int_x W_k(\mathbf{x}) q_s(\mathbf{x}, y) d\mathbf{x} = q_s(y = k)$$

*Referring to covariate shift assumption where $q_s(y|\mathbf{x}) = q_t(y|\mathbf{x})$ and $q_s(y) = q_t(y)$, we can deduce:*

$$\int_x W_k(\mathbf{x}) q_t(\mathbf{x}, y) d\mathbf{x} = q_t(y = k)$$

*Which means $\mathbb{E}_{q_t(\mathbf{x}, y)}[W_k(\mathbf{x})] = q_t(y = k)$, and as $\hat{W}_k(\cdot; \cdot)$ is equal to $W_k(\cdot)$, empirically we can show $\sum_{\mathbf{x}_i^c \in \mathbb{C}} \hat{W}_k(\mathbf{x}_i; w^*) = q_t(y = k)$. In this problem setting, we assume $q_s(y)$ is known that is equal to $q_t(y)$. Therefore, $w^*$ can be found easily by minimizing $\left( \sum_{k=1}^{K} \sum_{i=1}^{L} \hat{W}_k(\mathbf{x}_i; w) - q_s(y = k) \right)^2$.*

# B EXPERIMENTAL SETUPS

## B.1 BASELINES

- *Temperature Scaling* (Guo et al. (2017)): It is explained in Sec. (3.3)
- *Matrix and Vector Scaling* (Platt et al. (1999)): Matrix Scaling applies a linear transformation on the logits to soften them:

$$S_{y=\hat{y}_i}(\mathbf{x}_i; \theta, \mathbf{b}) = \max_k \sigma(\theta.\mathbf{f}(\mathbf{x}_i) + \boldsymbol{b})^{(k)}$$
$$\hat{y}_i = \arg\max_k \sigma(\theta.\mathbf{f}(\mathbf{x}_i) + \boldsymbol{b})^{(k)} \tag{9}$$

Where $\sigma$ is the softmax function which takes logit layer $\mathbf{f}(\mathbf{x})$ as an input. The parameters $\theta_{K \times K}$ and $\mathbf{b}_K$ are optimized with respect to NLL on the validation set. Vector Scaling is the relaxed version of Matrix Scaling in which $\theta_{K \times K}$ is a diagonal matrix.

- *ll-Dropout* Monte-Carlo Dropout(Gal & Ghahramani (2016)), A pre-trained model which is trained with dropout rate $p = 0.5$ only on the activation function before the last layer, and keeping the dropout active during the test with the same rate.
- *Ensembles* Ensembles of 10 networks trained independently on the entire dataset using the random initialization(Lakshminarayanan et al. (2017)).
- *SVI* Stochastic Variational Bayesian Inference for deep learning (Blundell et al. (2015); Wen et al. (2018) with the specific settings of training mentioned in (Ovadia et al. (2019))

## B.2 DATASETS

We apply the calibration method on different image classification datasets. For each experiment, the size of validation set is 20% of the test set which is selected randomly. For all the model-dataset we have trained them on the training set.

1. *CIFAR-10* (Krizhevsky et al. (2009)): It contains 60000, $32 \times 32$ color images of 10 different objects, with 6000 images per class. The size of training and test sets are 50000 and 10000 respectively.
2. *CIFAR-100* (Krizhevsky et al. (2009)): With the same setting as CIFAR-10, except it has 100 classes of different objects containing 600 images in each class.
3. *SVHN* (Netzer et al. (2011)): It contains $32 \times 32$ color images of numbers between 0 to 9 that has 73257 digits for training, 26032 digits for testing.
4. *MNIST* (LeCun et al. (1998b)): It contains $28 \times 28$ gray-scale images of numbers between 0 to 9. It has 60,000 images for training, and 10,000 images for test.
5. *Calthec-UCSD Birds* (Wah et al. (2011)): It contains 11,788 color images of 200 different birds species. We divided randomly into 7073 training, and 4715 testing samples.

## B.3 CALIBRATION METRICS

*Expected Calibration Error (ECE)*
ECE (Naeini et al. (2015)) measures the average gap between the accuracy and predicted probabilities. Based on this definition of calibration, ECE is proposed as empirical expectation error between the accuracy and confidence. It is calculated by partitioning the range of confidence between $[0, 1]$ into $B$ equally-spaced confidence bins and then assign the samples to each bin $B_b$ where $b = \{1, \ldots, B\}$ by their confidence range. Later it calculates the weighted absolute difference between the accuracy and confidence for each subset $B_l$. More specifically:

$$\text{ECE} = \sum_{b=1}^{B} \frac{|B_b|}{N} \left| \text{acc}(B_b) - \text{conf}(B_b) \right|, \tag{10}$$

where $N$ is the total number of samples. In this paper, we consider $B = 15$ to report the ECE error. ECE is not derivable function. Therefore mostly it is ignored as the loss function of the

post-processing approaches in calibrating the model by gradient decent optimizing methods.

*Brier Score*
Brier Score (Brier (1950)) is a scoring rule for measuring the accuracy of predicted probabilities. It is computed as the square error of predicted probability and the one-hot encoding of the correct label. That is:

$$B(\mathbf{x}_i, y_i) = \frac{1}{K} \sum_{y=1}^{K} \left(S_y(\mathbf{x}_i) - \delta(y - y_i)\right)^2 \tag{11}$$

## C  MORE EXPERIMENTAL RESULTS

### C.1  TABLES OF ACCURACY, NLL, ECE AND BRIER SCORE

We report additional results of the experiment applied in Sec. 5.1 to evaluate the behavior of UTS in calibration with the same training and test domains. We report the accuracy, NLL, ECE and Brier Score in Table 2, 3, 4, and 5, respectively. Notice that TS family approaches keep the accuracy unchanged while Matrix and Vector Scaling can change the accuracy. In two cases Matrix and Vector Scaling improve the accuracy. However, they get overfitted on validation set and lose the accuracy and calibration in general. We report the results for different calibration score to show that UTS will calibrate the model regarding different calibration evaluation metrics. NLL and Brier Score have related definition of calibration as both of them are proper scoring rule. But ECE has different calibration definition. The detail explanation are given in Sec. (B.3) for each score. We can see in Table 4, UTS even can calibrate the model better than TS for two dataset-model combination with ECE definition of calibration.

Table 2: Accuracy of different post-processing approaches for different model-datasets

| Dataset | Model | Uncalibrated, TS, UTS | Vector Scaling | Matrix Scaling |
|---|---|---|---|---|
| Birds | ResNet50 | **76.03 +/- 0.17** | 75.94 +/- 0.31 | 38.82 +/- 1.92 |
| MNIST | Lenet-5 | **99.19 +/- 0.03** | 99.18 +/- 0.03 | 99.11 +/- 0.05 |
| SVHN | ResNet110 | **96.09 +/- 0.08** | 96.07 +/- 0.08 | 96.08 +/- 0.08 |
| SVHN | DenseNet100 | 95.77 +/- 0.03 | **95.94 +/- 0.02** | 95.86 +/- 0.02 |
| CIFAR10 | DenseNet40 | **92.67 +/- 0.12** | 92.20 +/- 0.10 | 92.55 +/- 0.04 |
| CIFAR10 | DenseNet100 | **95.08 +/- 0.09** | 94.81 +/- 0.12 | 95.02 +/- 0.11 |
| CIFAR10 | ResNet110 | **93.62 +/- 0.13** | 93.53 +/- 0.14 | 93.49 +/- 0.13 |
| CIFAR10 | VGG16 | 93.42 +/- 0.12 | 93.39 +/- 0.14 | **93.40 +/- 0.11** |
| CIFAR100 | DenseNet40 | **71.56 +/- 0.18** | 66.89 +/- 0.56 | 63.79 +/- 0.81 |
| CIFAR100 | DenseNet100 | **75.87 +/- 0.15** | 73.38 +/- 0.25 | 73.12 +/- 0.15 |
| CIFAR100 | ResNet110 | **70.41 +/- 0.26** | 67.24 +/- 0.20 | 65.27 +/- 0.28 |
| CIFAR100 | WideResNet40-4 | **79.80 +/- 0.22** | 79.57 +/- 0.15 | 79.79 +/- 0.17 |

Table 3: NLL↓ for UTS and other post-processing approaches. The mean and std are reported in 20 independent runs.

| Dataset | Model | Uncalibrated | TS | UTS | Vector Scaling | Matrix Scaling |
|---|---|---|---|---|---|---|
| Birds | ResNet50 | 0.9383 +/- 0.0086 | **0.9287 +/- 0.0077** | 0.9313 +/- 0.0078 | 0.9355 +/- 0.0093 | 7.1423 +/- 0.2424 |
| MNIST | Lenet-5 | 0.1044 +/- 0.0064 | **0.0243 +/- 0.0013** | 0.0429 +/- 0.0009 | 0.0988 +/- 0.0073 | 0.1187 +/- 0.0094 |
| SVHN | ResNet110 | 0.2107 +/- 0.0035 | **0.1534 +/- 0.0023** | 0.1566 +/- 0.0030 | 0.1936 +/- 0.0035 | 0.2070 +/- 0.0036 |
| SVHN | DenseNet100 | 0.1803 +/- 0.0017 | **0.1608 +/- 0.0013** | 0.1735 +/- 0.0048 | 0.1687 +/- 0.0025 | 0.1754 +/- 0.0017 |
| CIFAR10 | DenseNet40 | 0.2895 +/- 0.0058 | **0.2221 +/- 0.0034** | 0.2848 +/- 0.0419 | 0.2912 +/- 0.0076 | 0.2883 +/- 0.0061 |
| CIFAR10 | DenseNet100 | 0.1973 +/- 0.0060 | **0.1559 +/- 0.0035** | 0.1635 +/- 0.0051 | 0.2006 +/- 0.0098 | 0.1965 +/- 0.0061 |
| CIFAR10 | ResNet110 | 0.3134 +/- 0.0083 | **0.2069 +/- 0.0047** | 0.2341 +/- 0.0109 | 0.2907 +/- 0.0090 | 0.3103 +/- 0.0081 |
| CIFAR10 | VGG16 | 0.2608 +/- 0.0060 | **0.2036 +/- 0.0039** | 0.2099 +/- 0.0065 | 0.2503 +/- 0.0093 | 0.2573 +/- 0.0064 |
| CIFAR100 | DenseNet40 | 1.0964 +/- 0.0067 | **1.0064 +/- 0.0053** | 1.0217 +/- 0.0053 | 1.2876 +/- 0.0215 | 1.4914 +/- 0.0520 |
| CIFAR100 | DenseNet100 | 1.1285 +/- 0.0072 | **0.8751 +/- 0.0042** | 0.8889 +/- 0.0054 | 1.2470 +/- 0.0186 | 1.2761 +/- 0.0129 |
| CIFAR100 | ResNet110 | 1.2442 +/- 0.0132 | **1.0466 +/- 0.0090** | 1.0636 +/- 0.0140 | 1.3869 +/- 0.0186 | 1.5358 +/- 0.0204 |
| CIFAR100 | WideResNet40-4 | 0.8751 +/- 0.0111 | **0.8192 +/- 0.0096** | 0.8439 +/- 0.0089 | 0.8757 +/- 0.0095 | 0.8609 +/- 0.0103 |

Table 4: ECE↓ for UTS and other post-processing approaches. The mean and std are reported in 20 independent runs.

| Dataset | Model | Uncalibrated | TS | UTS | VS | MS |
|---------|-------|--------------|-----|-----|-----|-----|
| Birds | ResNet50 | 0.0601 +/- 0.0015 | 0.0397 +/- 0.0025 | **0.0350 +/- 0.0024** | 0.0576 +/- 0.0019 | 0.2940 +/- 0.0089 |
| MNIST | Lenet-5 | 0.0074 +/- 0.0004 | 0.0027 +/- 0.0009 | **0.0199 +/- 0.0014** | 0.0074 +/- 0.0004 | 0.0077 +/- 0.0005 |
| SVHN | ResNet110 | 0.0266 +/- 0.0004 | **0.0044 +/- 0.0009** | 0.0127 +/- 0.0036 | 0.0247 +/- 0.0003 | 0.0264 +/- 0.0005 |
| SVHN | DenseNet100 | 0.0160 +/- 0.0004 | **0.0059 +/- 0.0009** | 0.0106 +/- 0.0013 | 0.0139 +/- 0.0003 | 0.0156 +/- 0.0005 |
| CIFAR10 | DenseNet40 | 0.0409 +/- 0.0007 | **0.0070 +/- 0.0009** | 0.0351 +/- 0.0039 | 0.0410 +/- 0.0021 | 0.0419 +/- 0.0008 |
| CIFAR10 | DenseNet100 | 0.0262 +/- 0.0010 | **0.0073 +/- 0.0017** | 0.0157 +/- 0.0040 | 0.0281 +/- 0.0033 | 0.0262 +/- 0.0017 |
| CIFAR10 | ResNet110 | 0.0431 +/- 0.0009 | **0.0088 +/- 0.0009** | 0.0226 +/- 0.0021 | 0.0435 +/- 0.0023 | 0.0439 +/- 0.0012 |
| CIFAR10 | VGG16 | 0.0419 +/- 0.0010 | **0.0169 +/- 0.0010** | 0.0297 +/- 0.0043 | 0.0402 +/- 0.0011 | 0.0419 +/- 0.0011 |
| CIFAR100 | DenseNet40 | 0.0861 +/- 0.0011 | **0.0109 +/- 0.0018** | 0.0390 +/- 0.0062 | 0.1312 +/- 0.0054 | 0.1553 +/- 0.0102 |
| CIFAR100 | DenseNet100 | 0.1223 +/- 0.0016 | **0.0152 +/- 0.0011** | 0.0372 +/- 0.0063 | 0.1416 +/- 0.0020 | 0.1452 +/- 0.0039 |
| CIFAR100 | ResNet110 | 0.1265 +/- 0.0006 | **0.0102 +/- 0.0027** | 0.0447 +/- 0.0048 | 0.1567 +/- 0.0045 | 0.1740 +/- 0.0074 |
| CIFAR100 | WideResNet40-4 | 0.0894 +/- 0.0012 | **0.0402 +/- 0.0027** | 0.0689 +/- 0.0033 | 0.0839 +/- 0.0037 | 0.0850 +/- 0.0023 |

Table 5: Brier Score↓ for UTS and other post-processing approaches. The mean and std are reported in 20 independent runs.

| Dataset | Model | Uncalibrated | TS | UTS | VS | MS |
|---------|-------|--------------|-----|-----|-----|-----|
| Birds | ResNet50 | **0.0017 +/- 0.0000** | **0.0017 +/- 0.0000** | **0.0017 +/- 0.0000** | **0.0017 +/- 0.0000** | 0.0043 +/- 0.0001 |
| MNIST | Lenet-5 | 0.0015 +/- 0.0001 | **0.0012 +/- 0.0001** | 0.0014 +/- 0.0000 | 0.0015 +/- 0.0001 | 0.0016 +/- 0.0001 |
| SVHN | ResNet110 | 0.0066 +/- 0.0001 | **0.0061 +/- 0.0001** | 0.0062 +/- 0.0001 | 0.0065 +/- 0.0001 | 0.0066 +/- 0.0001 |
| SVHN | DenseNet100 | 0.0066 +/- 0.0001 | 0.0065 +/- 0.0001 | 0.0065 +/- 0.0001 | **0.0063 +/- 0.0001** | 0.0064 +/- 0.0001 |
| CIFAR10 | DenseNet40 | 0.0118 +/- 0.0001 | 0.0109 +/- 0.0001 | 0.0115 +/- 0.0002 | 0.0121 +/- 0.0002 | 0.0118 +/- 0.0001 |
| CIFAR10 | DenseNet100 | 0.0080 +/- 0.0002 | **0.0075 +/- 0.0002** | 0.0076 +/- 0.0002 | 0.0084 +/- 0.0005 | 0.0080 +/- 0.0003 |
| CIFAR10 | ResNet110 | 0.0108 +/- 0.0002 | **0.0098 +/- 0.0001** | 0.0100 +/- 0.0001 | 0.0110 +/- 0.0003 | 0.0108 +/- 0.0002 |
| CIFAR10 | VGG16 | 0.0108 +/- 0.0002 | **0.0098 +/- 0.0003** | 0.0100 +/- 0.0001 | 0.0106 +/- 0.0002 | 0.0107 +/- 0.0002 |
| CIFAR100 | DenseNet40 | 0.0040 +/- 0.0000 | **0.0039 +/- 0.0000** | **0.0039 +/- 0.0000** | 0.0048 +/- 0.0001 | 0.0052 +/- 0.0002 |
| CIFAR100 | DenseNet100 | 0.0037 +/- 0.0000 | **0.0033 +/- 0.0000** | 0.0034 +/- 0.0000 | 0.0040 +/- 0.0000 | 0.0040 +/- 0.0001 |
| CIFAR100 | ResNet110 | 0.0043 +/- 0.0000 | **0.0040 +/- 0.0000** | **0.0040 +/- 0.0000** | 0.0048 +/- 0.0001 | 0.0051 +/- 0.0001 |
| CIFAR100 | WideResNet40-4 | 0.0030 +/- 0.0000 | **0.0029 +/- 0.0000** | 0.0030 +/- 0.0000 | 0.0031 +/- 0.0000 | 0.0030 +/- 0.0000 |

## C.2 COMPARING DIFFERENT TS FAMILY APPROACHES TO UTS FOR DOMAIN SHIFT SCENARIOS

In this section, we provide more results to compare the behavior of UTS with TS and TS-Target in different domain shift applied to CIFAR10 dataset. Details of shifting operation can be found in (Hendrycks & Dietterich, 2019). We can see in Fig. 5 that TS-Target has the most calibration robustness to the domain shift and followed by UTS with small gap.

## C.3 MORE EXPERIMENTAL RESULTS FOR SENSITIVITY OF TS TO THE LABELING NOISE

In this section, we provide more experiments for more models and datasets. As we can see in Fig. 3, TS shows a huge sensitivity to the noise of labeling in calibration set. UTS is completely robust to this noise as it is an unsupervised method. It shows if TS wants to calibrate the model without available labeled data, the labeling phase should be handled precisely. Otherwise TS results are not reliable.

## C.4 ANALYSIS OF UTS SENSITIVITY TO NUMBER OF SAMPLES

In this experiment, we show the impact of number of available calibration samples on TS, UTS and TS-Target. As we can see in Fig.7, TS and UTS vary significantly when the number of samples are really few (between $30 \sim 50$) while TS-Target is not affected severely (the variance of TS-Target comparing to other methods is small then it is not properly shown in the image). However, by increasing the number of samples to 500, UTS reaches to the optimal $T^*_{UTS}$ and after that increasing the number of samples does not consequences better results.

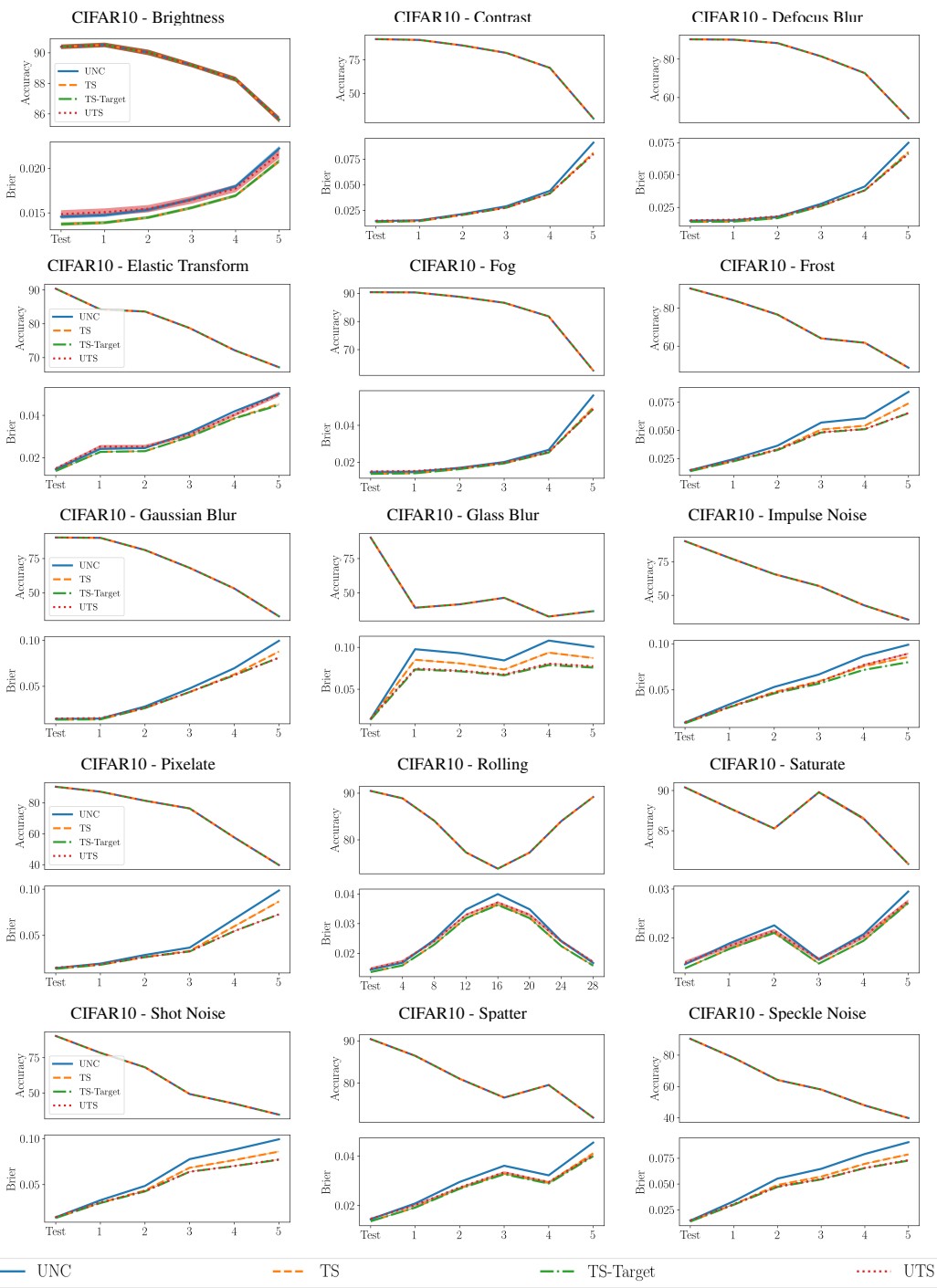

Figure 5: Robustness of UTS to domain shift problem in compare to TS, Ts-Target and uncalibrated methods. UTS with small gap caan follow TS-Target which shows it is a robust method to domain shift

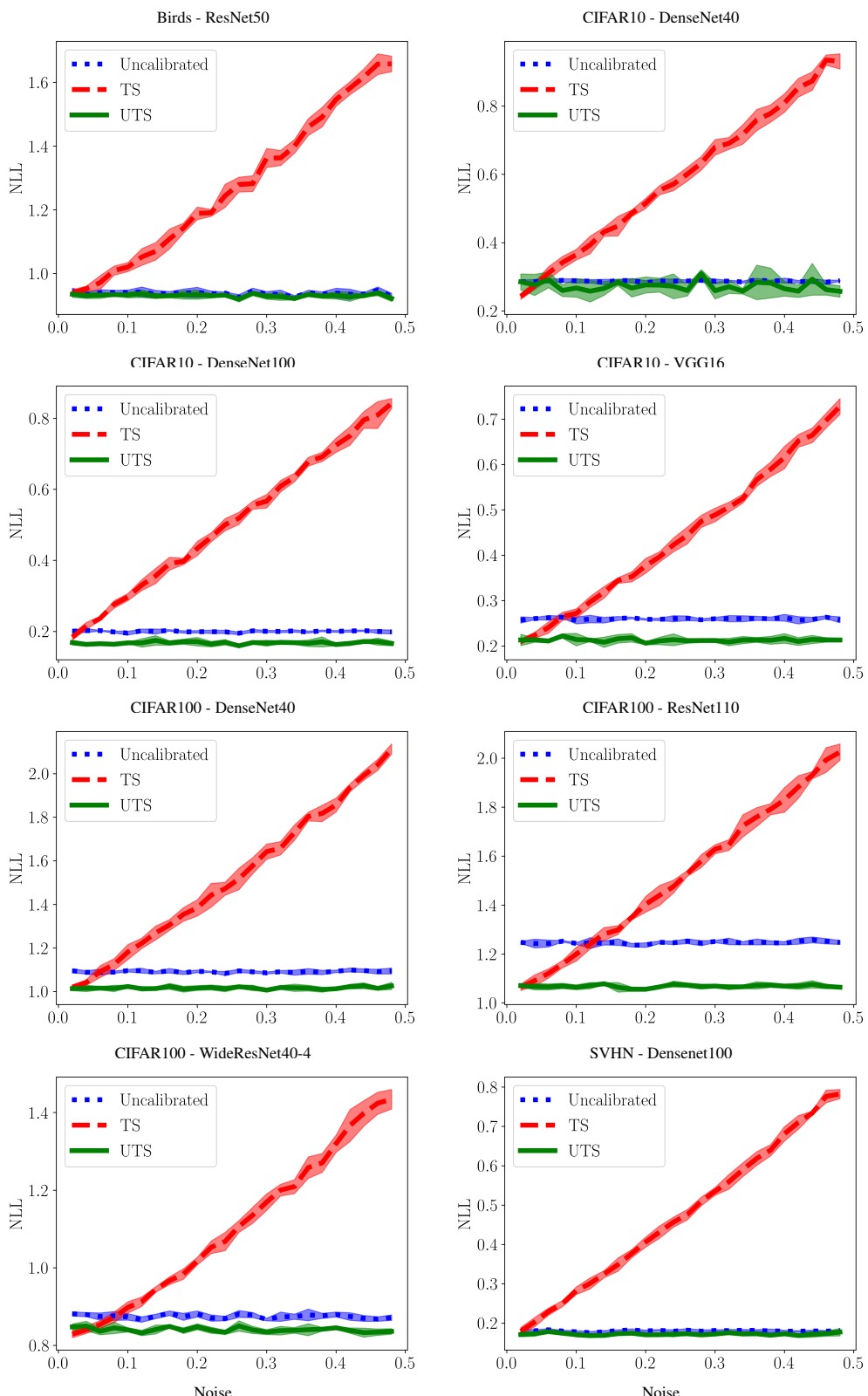

Figure 6: Sensitivity of TS method to the labeling noise in calibration set in compare to UTS and uncalibrated methods. UTS is not sensitive to the labeling noise in calibration sets at all as it is an unsupervised approach. Shaded regions represent std over 20 runs.

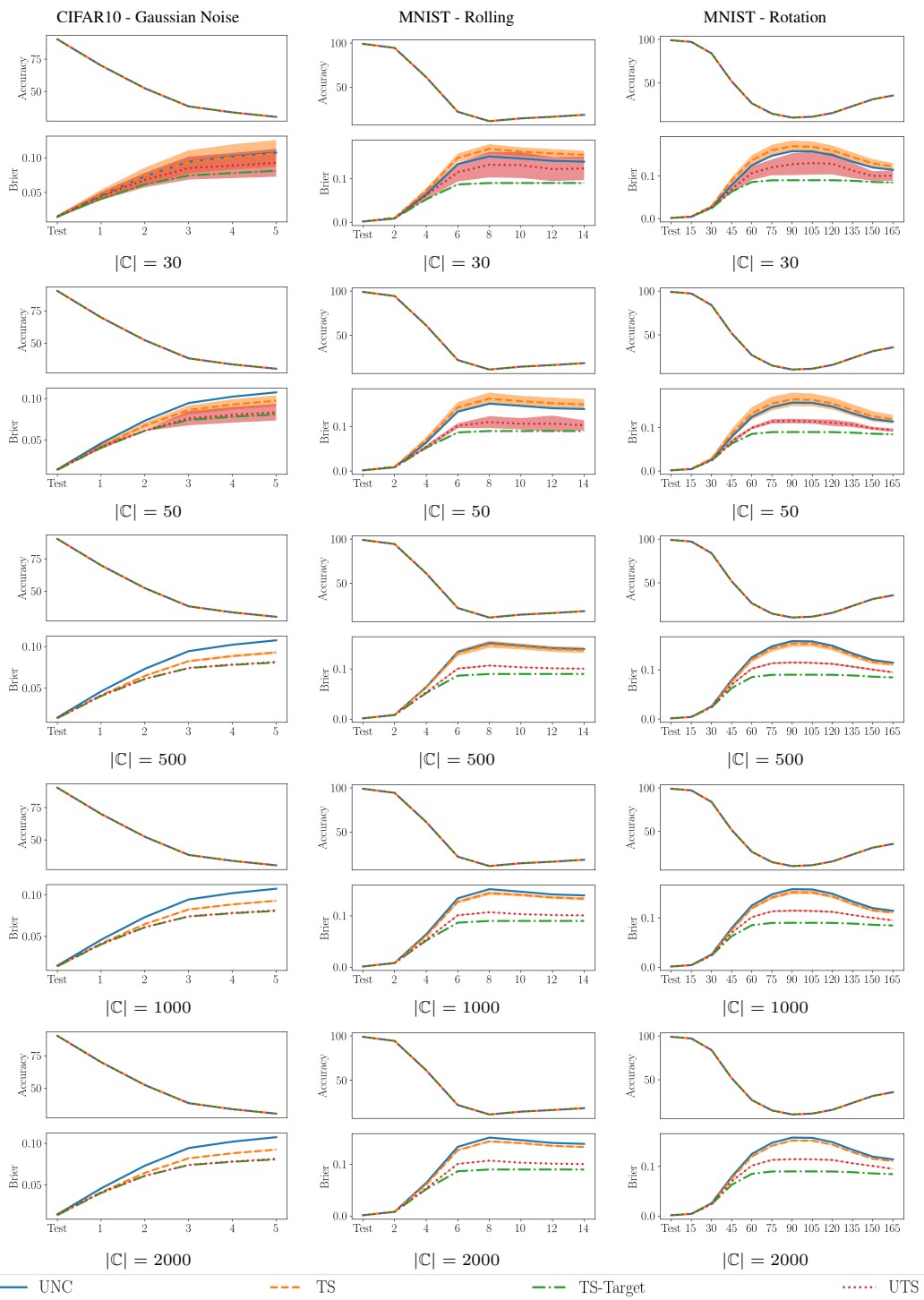

Figure 7: Comparing the stability of different TS family approaches to the number of calibration samples. TS-Target has the best stability. However UTS converges quickly to the optimal solution. Consider that collecting samples for UTS comparing to TS-Target is much less expensive task as it does not need labeled samples. The number of samples in calibration set $\mathbb{C}$ from top row to the bottom is 30, 50, 500, 1000, and 2000.

