# OpenReview forum: "Unsupervised Temperature Scaling: Robust Post-processing Calibration for Domain Shift"
_ICLR.cc/2020/Conference — Reject_

### Official Review · AnonReviewer2 · 2019-10-19
**Official Blind Review #2**

**Rating:** 3

**Review:**

I've read the rebuttal and unfortunately, I'd like to keep my score as is. I still think the assumption made in the paper is too limiting for most practical settings.

#########################

The paper proposes an unsupervised calibration method in a domain adaptation setting. The approach is based on the well known temperature scaling and does not require labels for the calibration set. The problem of calibration under domain shift is an important problem in areas where uncertainty estimation is useful; the paper tackles this problem and relaxes the assumption of knowing the input distribution. The method does not rely on the labels in the calibration set but has a major limitation of knowing the task distribution which may not be true in many practical settings where uncertainty estimation is relevant (such as medical diagnostics).

This assumption may not be a negative point for the paper as any domain adaptation problem needs at least some minimal assumptions; however,  the limits of the proposed method should be studied with respect to this assumption. For instance, in the experiments how robust are the experiments with respect to the assumption of a known q(y)? In the practical applications of the method in medical domain and self driving cars, q(y) is only known up to some approximation; so understanding the robustness of the method w.r.t. to this assumption is critical in real applications.

Also with the recent attention to calibration and uncertainty estimation in DL; I believe the acceptance bar for papers in this area has risen. Unfortunately, most papers in this area rely on completely synthetic experiments which makes their impact limited. I understand that ground truth uncertainty may not be available in some of these domains; however, other indirect metrics such as missclassification detection can be used. There are also medical datasets available (e.g. Diabetic Retinopathy) that can be used for evaluation.

To summarize, the paper addresses an important problem of calibration under domain shift but it needs some more empirical work to show the real advantage and limitations of the proposed method in a practical setting.

**Experience Assessment:**

I have published one or two papers in this area.

**Review Assessment: Checking Correctness Of Derivations And Theory:**

I assessed the sensibility of the derivations and theory.

**Review Assessment: Checking Correctness Of Experiments:**

I assessed the sensibility of the experiments.

**Review Assessment: Thoroughness In Paper Reading:**

I read the paper at least twice and used my best judgement in assessing the paper.

---

> ### Author Response · Authors · 2019-11-13
> **Reviewer #2 Reply**
>
> Thank you for raising the important point of applicability of the methods in the field of machine learning, which we think it was our concern to propose  UTS, too.
> The main advantage of UTS is ease of use and applicability in real scenarios. However, in this paper, we dedicate the main part of the paper to discuss the UTS assumptions and its proof of robustness to the domain shift as UTS is a completely new idea and needs justification.
>
> UTS is applicable to real scenarios as:
>
> 1- the time complexity of UTS is O(1) which makes it really appealing for the application.
> 2- it is a post-processing method, then it can be used to calibrate the model which is already trained toward the distribution of any new dataset.
>  3- it does not need labels for the samples to calibrate the model which makes it really useful in many applications.
>
> The main concern is about UTS assumption. It needs to know $q_t(y)$ to function properly. Considering a K class classification problem, y is a discrete random variable. Therefore, by having access to the samples from the training set, approximating $q_s(y)$ is just by computing empirically the ratio of each class number of samples to the total number of samples in the training set. As we consider Covariate shift assumption, then $q_s(y)= q_t(y)$ and simply we can approximate it for the test domain, too.
>
> Covariate Shift assumption that we made UTS based on, is the most famous domain adaptation assumption that is valid in many applications. When the training and test domains have a difference in representation distribution, the distribution shift scenario is categorized as Covariate Shift assumption. In real scenarios, It is very common that we train a model on a dataset but when we want to apply the model on the test samples, they have different illumination, background, resolution or viewpoint from the training one. This setting is categorized as covariate shift assumption. In this case, test data keeps the same label distribution and only the representation of samples is changed based on the domain shift. Even in medical applications, for instance in one region,  the rate of occurrence of different skin diseases is not changed but the condition in which the images are taken for skin disease detection could be changed from one healthcare center to the other. This causes the classifier trained on skin images from one center to drop the accuracy during the test phase on the other center and we need the accurate certainty adjustment to let the system make a decision about when the output is trustworthy or not.
>
> Update to the paper:
> As it seems this explanation is missing in the paper, we add more details to the Section 1, Introduction and we add a new paragraph in Section 4.1 to explain about the validity of UTS assumptions in the paper.
>
> We completely agree that it could be an interesting topic to see that UTS is how much robust to violating its assumptions in real situations. However, in this paper, we only open the door to the concept of certainty adjustment for domain shift and this can be considered as the future work.

---

### Official Review · AnonReviewer1 · 2019-10-20
**Official Blind Review #1**

**Rating:** 1

**Review:**

The authors present an algorithm for postprocessing neural networks to ensure calibration under domain shift.
Calibration under domain shift is an interesting challenge that has been receiving increasing attention and tackling this in an unsupervised manner is an interesting approach. However, I have 2 major concerns regarding the approach presented by the authors.

What makes calibration under domain shift useful and appealing is that the model is then robust against any changes in the test distribution that can occur during the life cycle of a model. These often include erroneous/samples (corresponding to truly OOD samples), but also gradual domain shift, where the test distribution continuously moves away from the training distribution (e.g. due to a continuous drift in user behaviour/change in customer base) or unforeseen changes. My first major concern is regarding the requirements for UTS, which render this approach not very useful in many of these practical  applications: UTS first requires knowledge of and access to the test distribution; in addition it assumes that the distribution of the labels remains unaffected under domain shift. These assumptions are violated in the practical applications described above, in particular those where a gradual, continuous domain shift occurs - in this case, access to the test distribution is difficult since it changes continuously. On this note I also would have liked to see some analysis on how performance depends on the number of samples that are available from the test set, since in practice this might be substantially smaller than the full test set used.
Furthermore, I find the assumption that the distribution of labels remains unchanged problematic (q_s(y) = q_t(y) and even q_s(y|x)=q_t(y|x)): once sufficiently out-of-domain, labels become meaningless and predictions for truly OOD samples should have maximum entropy. Even for small domain shifts in practical applications it is not clear why q_s(y|x)=q_t(y|x) should hold and it would have been useful to see a discussion and some robustness analysis on this.
Finally,  the algorithm requires re-calibration whenever the test distribution changes, which in practice is  often not clear (and part of the reason why dealing with predictions under domain shift is so challenging).

In addition to doubts on practical applicability, my second major concern is regarding the depth of the evaluation.
First, while the authors present some comparisons to probabilistic methods, I am missing a crucial comparison to Evidential Deep Learning (Sensoy et al, NeurIPS 2018), which results in far superior performance than deep ensembles, SVI or dropout. Importantly, the comparisons to probabilistic approaches presented by the authors are very limited. The big advantage of those approaches is that, once trained, no further recalibration is necessary and well calibrated predictions can be made for any level of domain shift, whereas UTS requires a recalibration step for very level of domain shift. That is why I think it is crucial to not only show one arbitrarily picked level of domain shift for each dataset/perturbation, but calibration across all levels of domain shift, as for TS and TS-Target; since no recalibration is required for those probabilistic approaches
 this is very straight-forward and would be very informative - especially since e.g Figure 5 shows that UTS has only very minor advantages over TS in many settings.
I appreciate that the authors report some performance in terms of ECE in the supplement, but I think it would be very informative to report performance in terms of ECE for all domain-shift experiments: The Brier score conflates accuracy with calibration (see eg the 2 component decomposition), whereas ECE directly quantifies calibration and is hence easier to interpret and arguably the more meaningful measure when quantifying calibration.

Minor:  I find the manuscript lacks clarity. Aspects such as the definition of calibration as well as implications and interpretation of Proposition 1 should be described in more detail in the manuscript.


**Experience Assessment:**

I have published one or two papers in this area.

**Review Assessment: Checking Correctness Of Derivations And Theory:**

I carefully checked the derivations and theory.

**Review Assessment: Checking Correctness Of Experiments:**

I carefully checked the experiments.

**Review Assessment: Thoroughness In Paper Reading:**

I read the paper thoroughly.

---

> ### Author Response · Authors · 2019-11-13
> **Reviewer #1 Reply**
>
> Thank Reviewer #1 for thoroughly reading the paper, comments, and discussions, however, there are important points that it seems emerging concerns. We try to clarify them by bringing detailed explanations one by one:

---

> > ### Author Response · Authors · 2019-11-13
> > **The first concern is about the UTS assumption,  robustness and applicability and why it is not a useful tool for OOD detection, label shift or concept drift problems.**
> >
> > When we tackle with domain shift problem, first we should clarify our assumptions and show for which type of problems we are targeting to propose a solution. UTS focus on one type of shifting problem which is called Covariate shift. Covariate Shift is the famous assumption (Discriminative learning under Covariate shift, Bickel et al., JMLR 2009) which is applied to the problem that distribution shift happens in the representation space and not in the label space. For instance in an image classification problem, if the distribution shift is caused by the difference in illumination, change of viewpoint, background or camera capturing resolution, etc between two domains, it is categorized as Covariate shift assumption. In this setting, the distribution of labels in the source and target is staying the same  $q_s(y) = q_t(y)$ while the representation distribution of input images differs $q_s(x)\neq q_t(x)$  and because both domains contains the same objects under different representation $q_s(y|x)=q_t(y|x)$ . In this condition, the classifier faces an accuracy drop in the target domain and we need to have access to the accurate certainty adjustment to help the system to make a decision based on.
> >
> > Covariate shift robustness and applicabilities are studied in the literature (Adaptive learning with covariate shift-detection for the motor imagery-based brain-computer interface, Raza et al., softcomputing 2016). Covariate shift can not address the other types of distribution shifts like label shift or OOD. Then it is reasonable that UTS cannot be a useful solution for them.
> >
> > UTS never assumes that it needs test distribution (referring to UTS assumption in Section 3.1, UTS objectives part). It only assumes that it knows label distribution of training samples (source domain) which is $q_s(y)$. In this paper, as we focus on classification problems, then we limit y to a discrete set of {1,..., K} which means K class. In this case, with balanced classes, $q_s(y)$ is easily considered as 1/K and for imbalance problems, as the class distribution is discrete, it can be easily estimated empirically by counting the number of each class samples with respect to the total number of samples in the training set. Under Covariate Shift assumption, $q_s(y) = q_t(y)$ which means class distribution is considered to stay the same between the source and target domains. Then after computing $q_s(y)$ it can be used as $q_t(y)$, to calibrate the model using the UTS method.
> >
> > However, our goal was not to propose a method for continuous domain shift problem, UTS definitely can be used for this purpose. If the domain shift happens during the time under Covariate shift assumption, as UTS uses few unlabeled data collected from the test domain and it is a light post-processing approach with time complexity O(1), it can easily adjust the uncertainty with high frequency that covers the distribution shift during the time. It only needs to know $q_s(y)$  which is assumed not changing and has access to a few numbers of samples from the test domain in each probable considering time slot that there is a chance of distribution shift. UTS does not have any tool to detect the distribution shift happened or not. But as it is a light calibration tool and it works in both conditions of with and without distribution shift then the user can frequently use UTS to calibrate the model to be sure that the model is calibrated.
> >
> > We have updated the paper and add some explanation about the UTS assumption in Section 1, in the Introduction part, one paragraph about the validity of UTS in practice to discuss more about Covariate shift and where this assumption is valid, and how to compute $q_s(y)$ in Section 4.1. We also make it more clear that UTS is not a tool for OOD and adversary detection in Section 2, “Robustness to the domain shift” part.
> >
> > 1- How many samples UTS needs to work properly?
> > As we mentioned in the paper, eventually, in all reported results, we select only 20% of the test dataset samples for adjusting the certainty for UTS and the rest of it for reporting the results. Then we already have considered the assumption of a few numbers of samples for adjusting calibration by UTS. But it would be a nice analysis to see the performance of UTS vs the number of available samples that we try to add it to the paper.

---

> > > ### Author Response · Authors · 2019-11-13
> > > **The second concern is about the depth of experiments.**
> > >
> > > 1-Why comparing with EDL is missing?
> > > UTS assumption is not based on detecting OOD and adversaries which EDL shows superior results than ensembles, SVI, and dropout in them. Therefore the comparison of UTS with EDL would be meaningless. To have reliable results, we use the benchmark designed specifically for domain shift concept in calibration research field (Can You Trust Your Model's Uncertainty? Evaluating Predictive Uncertainty Under Dataset Shift, ovaldia et al, NeurIPS2019)  to report the UTS behavior. All the selected baselines and shifting settings are based on this benchmark.
> > >
> > > 2- once you train a probabilistic model it can handle all types of domain shift, then what is the benefit of UTS?
> > > Ovaldia et al, only show the behavior of probabilistic calibration baselines in facing the domain shift problem and depict which one is more robust to the domain shift in compared to the others. They do not claim that the probabilistic approaches are designed to handle the domain shift problem.
> > > We compare UTS with probabilistic approaches only to show that it can achieve better calibration results under domain shift conditions, compared to the other calibration methods (probabilistic approaches) that actually are not designed for the domain shift. There is no specifically designed robust calibration method for domain shift problem that we can compare UTS with.
> > >
> > > Also, probabilistic approaches suffer from huge training time and complicated parameter fine-tuning to get good results. They also have no calibration solution for the model that is already trained and it is not calibrated. In the case of ensemble, however, the training is simple, it suffers from the need for fast accessible big memory to keep the trained deep models in it in the inference phase. Then in many applications such as mobile applications, it is not easy to use them.
> > >
> > > 3- To have a fair comparison, report the results with ECE?
> > > About measuring the calibration error, we should clarify that all three main calibration measures NLL, Brier Score, and ECE suffer from the influence of accuracy on their results. In ECE, as the deep networks are likely to be overconfident which means it is classifying the samples with high confidence, most of the samples are fallen in the high confidence bin (0.8~1). Then, when the model is more accurate the difference between confidence and accuracy will be decreased in this bin that has the majority weight in calculating ECE. Therefore,  ECE shows a lower error rate. ECE is also a problematic calibration measure that recently in the literature people pay attention to them (Measuring Calibration in Deep Learning, Nixon et al, CVPRW 2019).
> > >
> > > 4- why did you select arbitrary domain shift points to report the results of UTS in comparing to other probabilistic approaches?
> > > This fact is true that the more accurate networks are more calibrated. But here we do not want to improve the calibration by improving the accuracy and we want to show the impact of UTS in adjusting the probability output of the softmax layer to improve the calibration. Then selecting the points to report the calibration error in Figure 3 is not an arbitrary selection. We apply different levels of domain shift and select the point that nearly all the methods have similar accuracy on that point (the vertical axis of Figure 3 shows accuracy), then report the calibration results for that to have a fair comparison. TS, TS-Target, and UTS do not change the accuracy of the uncalibrated model. Then if we do not select the points, we compare them with probabilistic approaches that all of them improve the accuracy of the model and obtain higher accuracy than the uncalibrated model for different degrees of domain shift. Then, the impact of better accuracy would cover the impact of robustness to domain shift for comparison and the comparing becomes meaningless. In Figure 2&5, as TS, UTS, and TS-Target have the same accuracy as the uncalibrated model then we can compare them for all degree of domain shift.
> > >
> > > We should also clarify that the Brier Score error range is small. Then the minor improvement in comparing TS to UTS in Figure 5 is actually significant.

---

> > > > ### Author Response · Authors · 2019-11-13
> > > > **Minor Concerns**
> > > >
> > > > We address reviewer concerns by updating the text of the paper for proposition 1.

---

> > > > ### Comment · AnonReviewer1 · 2019-11-14
> > > > **Response to authors**
> > > >
> > > > I don't follow the authors' argumentation: as you state, EDL has a better perfromance than the probabilistic baseline models you compared to  - so surely this means it should be one of  the evaluated baselines. I think this is particularly important since, as the authors stated, it does not suffer from from the problem aof being a complex Bayesian neural network requiering huge training time and fine tuning (no more hyperparameters than a vanilla network).
> > > >
> > > > It is true that Brier score as well as ECE measure slightly differnt things - one of the advantages of the widely used ECE is that it does not conflate accuracy and calibration, which is why I still think this is an important compimentary metric that should be reported throughout.

---

> > > > > ### Author Response · Authors · 2019-11-15
> > > > > **Comparing UTS with EDL**
> > > > >
> > > > > As our arguments were not clear enough, we try to address them more properly again.
> > > > >
> > > > > Briefly, we have two main messages in this paper:
> > > > > 1- UTS is a post-processing approach with lightweight computational loss function that can calibrate already trained off the shelf models using few number of test samples.
> > > > > To address the first message that is the possibility of calibration without labels, we compare UTS with 3 other post processing approaches (Table 1) and show that the uncalibrated model can get more calibrated after applying UTS.
> > > > >
> > > > > 2- As UTS uses the test samples it can be robust to distribution shift between the test and training sets happens under Covariate shift setting.
> > > > > To address the second message, we compared UTS with different post-processing and probabilistic approaches to show that UTS can obtain better results for domain shift setting. In the literature there is no proposed approach that is designed for calibration under covariate shift assumption. Then we select these approaches and report the results of them comparing to UTS to show that UTS and TS-Target that are designed specifically for the domain shift under Covariate shift assumption can improve calibration better than the other 5 baselines that are not designed for domain shift.
> > > > >
> > > > > The message of the paper is not that UTS is better calibration method compared to the probabilistic approaches, or it is a solution for OOD detection or adversaries catching. Then we limit the comparison to these 8 baselines in the paper. There are several approaches including EDL, in the literature that already passed ensemble and dropout calibration error rate, but as our main message is not improving the calibration error rate of probabilistic approaches, we didn’t compare to them.
> > > > >
> > > > > I completely agree that showing more comparisons, brings more value to the paper and we will consider to add comparison to EDL also to the paper in the future.

---

> > > > > ### Author Response · Authors · 2019-11-15
> > > > > **Conflating accuracy with confidence in calibration metrics.**
> > > > >
> > > > > Reviewer Concern: ECE does not conflate accuracy and calibration then it can be used as the solution for Figure 3 experiment to show the UTS behavior for all the domain shifts degrees compared to probabilistic approaches , like Figure 2.
> > > > >
> > > > > Reply: Let us disagree with you in the concept of ECE. ECE like NLL and Brier conflates accuracy with calibration. As ECE is the average difference between the accuracy and confidence, and DNNs are overconfident (means their average confidence is high), then when the accuracy increases, the distance between average accuracy and average confidence will decrease and ECE  shows the model gets more calibrated. However, this improvement is only the impact of better accuracy.
> > > > >
> > > > > In all the papers that they report accuracy and ECE for different model-datasets, the inverse relation between ECE and accuracy can be recognized. Like the paper (on calibration of modern neural networks, Guo et al, ICML2017) for Table 1&Table S2 and (Can you trust your model's uncertainty? Evaluating predictive uncertainty under dataset shift, ovadia et al, Neurips 2019) for Figure3 &Figure S4~6 and also our manuscript for Table 2 &Table 4).
> > > > >
> > > > > Before reporting the results in this paper, we implemented different domain-shift-degree results (like figure 2) for the baselines in Figure 3, with calibration metrics ECE, NLL and Brier Score. All the different metrics results were aligned with each other and show better calibration error rates for the models that have better accuracy.
> > > > > As probabilistic approaches are more robust to accuracy drop in domain shifts and TS family approaches like UTS are not designed to improve the accuracy and only can adjust the confidence of the model without improving the accuracy. Then, they always were worse in calibration error rate in comparison to the probabilistic approaches. But the reason was not that UTS and TS-Target cannot calibrate the models. The reason was better accuracy of the other models for different degree of the domain shifts and sensitivity of calibration metrics to the accuracy level. Then we select the points that models have similar accuracy rate and report the results of calibration for them. When the accuracy of different baselines are similar to each other, TS-Target and UTS show better calibration results for domain shift which support the idea their confidence robustness to the domain shift.
> > > > >
> > > > > We think none of the metrics of Brier, ECE and NLL can address the problem of conflating accuracy with confidence. Then selecting the points in which models have almost similar accuracy range,  seems to be the best solution for  the challenge of reporting the results.

---

### Official Review · AnonReviewer3 · 2019-10-26
**Official Blind Review #3**

**Rating:** 6

**Review:**

The authors propose an approach for calibrated predictions under domain shift scenarios. The approach, that leverages (unlabeled) test samples allows for unsupervised post-processing calibration, even for off-the-shelf models for which the training data is not available. Experiments compare the proposed approach with existing calibration methods in shifted domains.

Equation (5) is confusing. If I understand correctly, the authors are simply making the point that q(x,y=k) can be written in terms of q(x,y\neq k) by weighting by the ratio of conditionals, which are available.

Sensitivity to noisy labels. The experiment is reasonable and the results are convincing, however, the authors do not justify why accurate (manual) labels on the target set are not feasible in many applications. The authors could point to a few examples for context.

The authors assume that q_s(y) = q_t(y), which seems restrictive in practice. Though it does not impact my opinion of the proposed approach, it seems narrow to think of a practical situation where the space of covariates is changing but the class composition remains unchanged. This is vaguely addressed in Section 6. Perhaps it can be elaborated further.

I enjoyed reading the paper, the proposed reinterpretation of NLL in terms of a weighted average and its approximation based on weights that do not depend on the labels but the (assumed known) labels marginal is interesting and seems to yield good results.

**Experience Assessment:**

I have published one or two papers in this area.

**Review Assessment: Checking Correctness Of Derivations And Theory:**

I assessed the sensibility of the derivations and theory.

**Review Assessment: Checking Correctness Of Experiments:**

I carefully checked the experiments.

**Review Assessment: Thoroughness In Paper Reading:**

I read the paper thoroughly.

---

> ### Author Response · Authors · 2019-11-13
> **Reviewer #3 Reply**
>
> Thank you for your insightful comments, and we are happy that you find the paper interesting. We address your concerns and add some parts to the paper accordingly:
>
> Concerns:
> 1- Equation (5) is confusing.
>
>  We mean exactly the point that the reviewer mentioned. As it was not clear in the text, we rewrite Eq.(5) explanation to make it more clear and precise.
>
> 2- the authors do not justify why accurate (manual) labels on the target set are not feasible in many applications. The authors could point to a few examples for context.
>
> We add three examples of applications (Neuron cells classification taken by electron microscope, pathology images and skin disease classification) that have expensive labeling procedure with high risk of labeling noise to the introduction of the paper (Section 1) to make it clear why labeling even for few number of samples is not possible sometimes.
>
> 3- The authors assume that $q_s(y) = q_t(y)$, which seems restrictive in practice.
>
> In domain shift, UTS is valid under Covariate Shift assumption for classification problem which means the test and training datasets are different in representation but keeps the same proportions of each class occurrence. Covariate shift assumption is a common domain adaptation assumption that is valid for many classification problems. For instance in medical image classification, it is very probable that the illumination, capturing noise, resolution, image size or viewpoint of the test images to be different from the training dataset. In this case,  the representation of two domains is changed  that means $q_s(x) \neq q_t(x)$ but the probability of happening a class of object is staying the same which means $q_s(y) = q_t(y)$.
> In classification problems as the $y$ domain is discrete, UTS only needs to calculate empirically the number of occurrence of each class to the total number of samples in the training set which is equal to $q_s(y)$ and use it as $q_t(y)$ to calibrate the model.
>
> Update to the paper:
> We add one extra paragraph to Section 4.1 with the title of  "Validity of UTS in Practice" focusing on Covariate shift assumption in practice and how to calculate $q_s(y)$  to address this important concern.

---

### Author Response · Authors · 2019-11-15
**Updates**

According to valuable reviewers’ concerns and suggestions, we add these parts to the paper:

1- Add complementary explanation to define the problem better (introduction, Section 1).
2- Add complementary explanation to highlight the difference between Covariate shift assumption considered in this paper and OOD and adversaries as other types of distribution shifts which are not the concern of this paper ( Section2 in “Robustness to the domain shift” part ).
3- Add one paragraph of discussion about the validity of  UTS in practice that discusses about the how to apply UTS in practice considering the assumptions (Section 4.1, validity of  UTS in practice).
4- Updating explanation of Eq.(1) and Eq.(5).
5- Add the results of sensitivity of UTS to the number of samples in Appendix  C.4, Figure 7. We did experiment for four levels of 30, 50, 500, 1000 and 2000 number of samples and show convergence behavior of UTS comparing to other TS family methods accordingly.

---

### Decision · Program_Chairs · 2019-12-19

**Decision:**

Reject

**Comment:**

The paper proposes a method called unsupervised temperature scaling (UTS) for improving calibration under domain shift.

The reviewers agree that this is an interesting research question, but raised concerns about clarity of the text, depth of the empirical evaluation, and validity of some of the assumptions. While the author rebuttal addressed some of these concerns, the reviewers felt that the current version of the paper is not ready for publication.

I encourage the authors to revise and resubmit to a different venue.